# Orthogonal Chaotic Binary Sequences Based on Bernoulli Map and Walsh Functions

**DOI:** 10.3390/e21100930

**Published:** 2019-09-24

**Authors:** Akio Tsuneda

**Affiliations:** Division of Informatics and Energy, Faculty of Advanced Science and Technology, Kumamoto University, Kumaomto 860-8555, Japan; tsuneda@cs.kumamoto-u.ac.jp; Tel.: +81-96-342-3853

**Keywords:** Bernoulli map, orthogonal sequence, Walsh functions, chaotic binary sequence, correlation property, i.i.d.

## Abstract

The statistical properties of chaotic binary sequences generated by the Bernoulli map and Walsh functions are discussed. The Walsh functions are based on a $2^{k}\times 2^{k}$ Hadamard matrix. For general *k* (=1,2,⋯), we will prove that $2^{k-1}$ Walsh functions can generate essentially different balanced and *i.i.d.* binary sequences that are orthogonal to each other.

## 1. Introduction

The simplest way to generate *chaos* is to use a one-dimensional(1D) nonlinear difference equation with a chaotic map. Chaotic sequences can be used as random numbers for several engineering applications, and there have been many works on chaos-based random number generation [1,2,3,4,5,6,7,8,9,10,11]. In general, *truly random numbers* should be a sequence of *i.i.d.* (independent and identically distributed) random variables with a uniform probability density, that is they give maximum entropy. Their typical model, for example, is a sequence obtained by trials of fair coin-tossing or dice-throwing. The design of many chaotic sequences of *i.i.d.* binary (or *p*-ary) random variables from a single chaotic real-valued sequence generated by a class of 1D nonlinear maps was established in [1,2,3], where it was shown that some symmetric binary (or *p*-ary) functions can produce *i.i.d.* binary (or *p*-ary) sequences if the map satisfies some symmetric properties.

In some engineering applications (e.g., communication, cryptography, the Monte Carlo method) of chaos-based random numbers, their statistical properties such as distributions and correlations are very important. Whereas there are some indices for defining chaos such as Lyapunov exponents, we concentrate on statistical properties in this paper. Thus, we discuss the statistical properties of orthogonal chaotic binary sequences generated by the Bernoulli map and Walsh functions based on Hadamard matrices, which was already discussed in [12]. As is well known, Walsh functions are the most famous orthogonal binary functions, and they can be applied to many applications (e.g., signal processing) [13,14,15,16,17]. In [12], we proved that the Bernoulli map and Walsh functions based on the $2^{k}\times 2^{k}$ (1≤k≤4) Hadamard matrix can generate $2^{k-1}$ different balanced *i.i.d.* binary sequences that are orthogonal to each other. Here, “balanced” means that the probability of “1” (or “0”) in the binary sequence is equal to 1/2. We conjectured that this holds for general positive integers *k*. In this paper, we will give a rigorous proof of this for general *k*(=1,2,⋯).

## 2. Preliminaries

For a nonlinear map τ(·), a *chaotic* sequence ${\left\{x_{n}\right\}}_{n=0}^{\infty }$ can be generated by a 1D difference equation:(1)$$x_{n+1}=\tau \left(x_{n}\right)\;\left(x_{n}\in I=\left[a,b\right],\;n=0,1,2,\cdots \right),$$
where $x_{n}=\tau^{n}\left(x\right)$ and $x=x_{0}$ is called an *initial value* or a *seed*. For an integrable function *G*, the average (expectation) of a sequence ${\left\{G\left(\tau^{n}\left(x\right)\right)\right\}}_{n=0}^{\infty }$ is defined by:(2)$$\langle G\rangle =\int_{I}G\left(x\right)f^{*}\left(x\right)dx,$$
which is very important in evaluating the statistics of chaotic sequences under the assumption that τ(·) is mixing on *I* with respect to an absolutely continuous invariant measure, denoted by $f^{*}\left(x\right)dx$.

**Definition** **1.**
*For two chaotic sequences ${\left\{G\left(\tau^{n}\left(x\right)\right)\right\}}_{n=0}^{\infty }$ and ${\left\{H\left(\tau^{n}\left(x\right)\right)\right\}}_{n=0}^{\infty }$ generated from a common seed x, their cross-correlation function is defined by:*
(3)$$C\left(\ell ;G,H\right)=\int_{I}\left(G\left(x\right)-\langle G\rangle \right)\left(H\left(\tau^{\ell }\left(x\right)\right)-\langle H\rangle \right)f^{*}\left(x\right)dx\;\left(\ell =0,1,2,\cdots \right),$$
*where ℓ is a time shift. If C(ℓ;G,H)=0, the two sequences ${\left\{G\left(\tau^{n}\left(x\right)\right)\right\}}_{n=0}^{\infty }$ and ${\left\{H\left(\tau^{n+\ell }\left(x\right)\right)\right\}}_{n=0}^{\infty }$ are called uncorrelated or orthogonal to each other. Note that C(ℓ;G,G) is the auto-correlation function of ${\left\{G\left(\tau^{n}\left(x\right)\right)\right\}}_{n=0}^{\infty }$.*


**Definition** **2.**
*The Perron-Frobenius (PF) operator $P_{\tau }$ of the map τ with an interval I=[a,b] is defined by:*
(4)$$P_{\tau }G\left(x\right)=\frac{d}{dx}\int_{\tau^{-1}\left(\left[a,x\right]\right)}G\left(y\right)dy$$
*which can be rewritten as:*
(5)$$P_{\tau }G\left(x\right)=\sum_{i}\left|g_{i}^{\prime }\left(x\right)\right|G\left(g_{i}\left(x\right)\right),$$
*where $g_{i}\left(x\right)$ is the i-th preimage of the map τ(·) [18].*


**Remark** **1.**
*The PF operator given in Definition 2 is very useful for evaluating correlation functions because it has the following important property [18]:*
(6)$$\int_{I}G\left(x\right)P_{\tau }\left\{H\left(x\right)\right\}dx=\int_{I}G\left(\tau \left(x\right)\right)H\left(x\right)dx.$$


**Remark** **2.**
*If:*
(7)$$P_{\tau }\left\{\left(G\left(x\right)-\langle G\rangle \right)f^{*}\left(x\right)\right\}=0,$$
*then we have:*
(8)C(ℓ;G,G)=C(ℓ;G,H)=0for ℓ≥1,
*which is obvious from Equations (Equation 3) and (Equation 6).*


**Remark** **3.**
*For a binary function B(x) (∈{0,1}), a sufficient condition for a binary sequence ${\left\{B\left(\tau^{n}\left(x\right)\right)\right\}}_{n=0}^{\infty }$ to be i.i.d. is given by [1]:*
(9)$$P_{\tau }\left\{\left(B\left(x\right)-\langle B\rangle \right)f^{*}\left(x\right)\right\}=0,$$
*which can also be expressed as:*
(10)$$P_{\tau }\left\{B\left(x\right)f^{*}\left(x\right)\right\}=\langle B\rangle f^{*}\left(x\right).$$


## 3. Hadamard Matrix and Walsh Functions

We introduce a $2^{k}\times 2^{k}$ Hadamard matrix $H_{k}$ defined by [13,14,15]:(11)$$\begin{array}{ccc} H_{k} & = & \left[\begin{array}{cc} H_{k-1} & H_{k-1}\\ H_{k-1} & -H_{k-1} \end{array}\right]\;\left(k=1,2,3,\cdots \right), \end{array}$$
(12)$$\begin{array}{ccc} H_{0} & = & \left[1\right] \end{array}$$
which is one of the orthogonal matrices whose rows (or columns) are orthogonal $2^{k}$-tuples. For example, $H_{3}$ is given by:(13)$$H_{3}=\left[\begin{array}{cccccccc} 1 & 1 & 1 & 1 & 1 & 1 & 1 & 1\\ 1 & -1 & 1 & -1 & 1 & -1 & 1 & -1\\ 1 & 1 & -1 & -1 & 1 & 1 & -1 & -1\\ 1 & -1 & -1 & 1 & 1 & -1 & -1 & 1\\ 1 & 1 & 1 & 1 & -1 & -1 & -1 & -1\\ 1 & -1 & 1 & -1 & -1 & 1 & -1 & 1\\ 1 & 1 & -1 & -1 & -1 & -1 & 1 & 1\\ 1 & -1 & -1 & 1 & -1 & 1 & 1 & -1 \end{array}\right].$$
Furthermore, $H_{k}$ can be expressed as:(14)$$H_{k}=\left[\begin{array}{cc} 1 & 1\\ 1 & -1 \end{array}\right]⊗H_{k-1}=H_{k-1}⊗\left[\begin{array}{cc} 1 & 1\\ 1 & -1 \end{array}\right],$$
where ⊗ denotes the Kronecker product.

Denote the (i,j)-th element of $H_{k}$ by $\left\{h_{i,j}^{\left(k\right)}\right\}$
$\left(i,j=0,1,\cdots ,2^{k}-1\right)$. Then, we consider binary functions $B_{i}^{\left(k\right)}\left(x\right)$ (∈{0,1}) defined by:(15)$$\begin{array}{ccc} B_{i}^{\left(k\right)}\left(x\right) & = & \sum_{j=0}^{2^{k}-1}\frac{1-h_{i,j}^{\left(k\right)}}{2}\left(\Theta_{\frac{j}{2^{k}}}\left(x\right)-\Theta_{\frac{j+1}{2^{k}}}\left(x\right)\right), \end{array}$$
(16)$$\begin{array}{ccc} \Theta_{t}\left(x\right) & = & \left\{\begin{array}{cc} 0 & \left(x<t\right)\\ 1 & (x\geq t). \end{array}\right. \end{array}$$
As an example, $B_{i}^{\left(3\right)}\left(x\right)$ (i=1,2,⋯,7) are shown in Figure 1. Note that $1-2B_{i}^{\left(k\right)}\left(x\right)$ (i=0,1,⋯) correspond to Walsh functions in natural (Hadamard) order, which include Rademacher functions [13,14].

**Proposition** **1.**
*The following relation:*
(17)$$B_{2i}^{\left(k\right)}\left(x\right)=B_{i}^{\left(k-1\right)}\left(x\right)\;\;\left(i=0,1,\cdots ,2^{k-1}-1\right)$$
*is satisfied. Namely, $B_{i}^{\left(k\right)}\left(x\right)$ ($i=0,1,\cdots ,2^{k}-1$) include all of $B_{i}^{\left(k-1\right)}\left(x\right)$ ($i=0,1,\cdots ,2^{k-1}-1$).*


**Proof.** From Equation (Equation 14), we have:
(18)$$h_{2i,2j}^{\left(k\right)}=h_{2i,2j+1}^{\left(k\right)}=h_{i,j}^{\left(k-1\right)}\;\;\left(i=0,1,\cdots ,2^{k-1}-1\right),$$
which leads us to obtaining:
$$\begin{array}{ccc} B_{2i}^{\left(k\right)}\left(x\right) & = & \sum_{j=0}^{2^{k}-1}\frac{1-h_{2i,j}^{\left(k\right)}}{2}\left(\Theta_{\frac{j}{2^{k}}}\left(x\right)-\Theta_{\frac{j+1}{2^{k}}}\left(x\right)\right)\\ & = & \sum_{j=0}^{2^{k-1}-1}\left\{\frac{1-h_{2i,2j}^{\left(k\right)}}{2}\left(\Theta_{\frac{2j}{2^{k}}}\left(x\right)-\Theta_{\frac{2j+1}{2^{k}}}\left(x\right)\right)+\frac{1-h_{2i,2j+1}^{\left(k\right)}}{2}\left(\Theta_{\frac{2j+1}{2^{k}}}\left(x\right)-\Theta_{\frac{2j+2}{2^{k}}}\left(x\right)\right)\right\}\\ & = & \sum_{j=0}^{2^{k-1}-1}\frac{1-h_{i,j}^{\left(k-1\right)}}{2}\left(\Theta_{\frac{2j}{2^{k}}}\left(x\right)-\Theta_{\frac{2j+2}{2^{k}}}\left(x\right)\right)\\ & = & \sum_{j=0}^{2^{k-1}-1}\frac{1-h_{i,j}^{\left(k-1\right)}}{2}\left(\Theta_{\frac{j}{2^{k-1}}}\left(x\right)-\Theta_{\frac{j+1}{2^{k-1}}}\left(x\right)\right)\\ & = & B_{i}^{\left(k-1\right)}\left(x\right). \end{array}$$This completes the proof. □

## 4. Orthogonal Chaotic Binary Sequences

For chaotic binary sequences ${\left\{B_{i}^{\left(k\right)}\left(\tau^{n}\left(x\right)\right)\right\}}_{n=0}^{\infty }$ ($i=1,2,\cdots ,2^{k}-1$) generated by a nonlinear map with I=[0,1] and $f^{*}\left(x\right)=1$, it is obvious that:(19)$$\langle B_{i}^{\left(k\right)}\rangle =\frac{1}{2},$$
that is, the binary sequences are *balanced*. Note that ${\left\{B_{0}^{\left(k\right)}\left(\tau^{n}\left(x\right)\right)\right\}}_{n=0}^{\infty }$ is excluded here since $B_{i}^{\left(k\right)}\left(x\right)\equiv 0$. Furthermore, we have:(20)$$\int_{I}B_{i}^{\left(k\right)}\left(x\right)B_{j}^{\left(k\right)}\left(x\right)dx=\frac{1}{4}\;\mathrm{for}\;i\neq j,$$
which gives:(21)$$C(0;B_{i},B_{j})=0\;\mathrm{for}\;i\neq j.$$
This implies that the binary sequences ${\left\{B_{i}^{\left(k\right)}\left(\tau^{n}\left(x\right)\right)\right\}}_{n=0}^{\infty }$ are *orthogonal* to each other.

In this paper, we employ Bernoulli map $\tau_{B}\left(x\right)$ defined by:(22)$$\tau_{B}\left(x\right)=\left\{\begin{array}{cc} 2x & \left(0\leq x<\frac{1}{2}\right)\\ 2x-1 & (\frac{1}{2}\leq x\leq 1), \end{array}\right.$$
which has the uniform invariant density $f^{*}\left(x\right)=1$ for the unit interval I=[0,1]. Figure 2 shows the map.

**Proposition** **2.**
*For Walsh functions $B_{i}^{\left(k\right)}\left(x\right)$ and Bernoulli map $\tau_{B}\left(x\right)$, the following relation:*
(23)$$B_{i}^{\left(k-1\right)}\left(\tau_{B}\left(x\right)\right)=B_{i}^{\left(k\right)}\left(x\right)\;\left(i=0,1,\cdots ,2^{k-1}-1\right)$$
*is satisfied.*


**Proof.** From Equation (Equation 14), we have:
(24)$$h_{i,j}^{\left(k-1\right)}=h_{i,j}^{\left(k\right)}=h_{i,j+2^{k-1}}^{\left(k\right)}\;\;\left(i,j=0,1,\cdots ,2^{k-1}-1\right).$$
Furthermore, for a threshold function $\Theta_{t}\left(x\right)$ and Bernoulli map $\tau_{B}\left(x\right)$, the following equation:
(25)$$\Theta_{t}\left(\tau_{B}\left(x\right)\right)=\Theta_{\frac{t}{2}}\left(x\right)-\Theta_{\frac{1}{2}}\left(x\right)+\Theta_{\frac{t}{2}+\frac{1}{2}}\left(x\right)$$
is satisfied as shown in Figure 2. Using Equations (Equation 15), (Equation 24), and (Equation 25), we have:
$$\begin{array}{ccc} B_{i}^{\left(k-1\right)}\left(\tau_{B}\left(x\right)\right) & = & \sum_{j=0}^{2^{k-1}-1}\frac{1-h_{i,j}^{\left(k-1\right)}}{2}\left(\Theta_{\frac{j}{2^{k-1}}}\left(\tau_{B}\left(x\right)\right)-\Theta_{\frac{j+1}{2^{k-1}}}\left(\tau_{B}\left(x\right)\right)\right)\\ & = & \sum_{j=0}^{2^{k-1}-1}\frac{1-h_{i,j}^{\left(k-1\right)}}{2}\left(\Theta_{\frac{j}{2^{k}}}\left(x\right)-\Theta_{\frac{j+1}{2^{k}}}\left(x\right)+\Theta_{\frac{j+2^{k-1}}{2^{k}}}\left(x\right)-\Theta_{\frac{j+1+2^{k-1}}{2^{k}}}\left(x\right)\right)\\ & = & \sum_{j=0}^{2^{k-1}-1}\frac{1-h_{i,j}^{\left(k-1\right)}}{2}\left(\Theta_{\frac{j}{2^{k}}}\left(x\right)-\Theta_{\frac{j+1}{2^{k}}}\left(x\right)\right)+\sum_{j=2^{k-1}}^{2^{k}-1}\frac{1-h_{i,j-2^{k-1}}^{\left(k-1\right)}}{2}\left(\Theta_{\frac{j}{2^{k}}}\left(x\right)-\Theta_{\frac{j+1}{2^{k}}}\left(x\right)\right)\\ & = & \sum_{j=0}^{2^{k}-1}\frac{1-h_{i,j}^{\left(k\right)}}{2}\left(\Theta_{\frac{j}{2^{k}}}\left(x\right)-\Theta_{\frac{j+1}{2^{k}}}\left(x\right)\right)\\ & = & B_{i}^{\left(k\right)}\left(x\right), \end{array}$$
which completes the proof. □

**Remark** **4.**
*From Propositions 1 and 2, we have:*
(26)$$B_{2i}^{\left(k\right)}\left(\tau_{B}\left(x\right)\right)=B_{i}^{\left(k-1\right)}\left(\tau_{B}\left(x\right)\right)=B_{i}^{\left(k\right)}\left(x\right)\;\left(i=1,2,\cdots ,2^{k-1}-1\right),$$
*which implies that some of the binary sequences ${\left\{B_{i}^{\left(k\right)}\left(\tau_{B}^{n}\left(x\right)\right)\right\}}_{n=0}^{\infty }$ ($i=1,2,\cdots ,2^{k}-1$) are time-shifted versions of others.*


**Theorem** **1.**
*For Bernoulli map $\tau_{B}\left(x\right)$, we have:*
(27)$$P_{\tau_{B}}\left\{B_{i}^{\left(k\right)}\left(x\right)-\langle B_{i}^{\left(k\right)}\rangle \right\}=0\;\;\left(i=2^{k-1},2^{k-1}+1,\cdots ,2^{k}-1\right).$$


**Proof.** Define ${\hat{\Theta }}_{t}\left(x\right)=\Theta_{t}\left(x\right)-\langle \Theta_{t}\rangle$. From [1], we have:
(28)$$P_{\tau_{B}}\left\{{\hat{\Theta }}_{t}\left(x\right)\right\}=\frac{1}{2}{\hat{\Theta }}_{\tau_{B}\left(t\right)}\left(x\right).$$
Furthermore, from Equation (Equation 14),
(29)$$h_{i,j+2^{k-1}}^{\left(k\right)}=-h_{i,j}^{\left(k\right)}=-h_{i,j}^{\left(k-1\right)}\;\;\left(j=0,1,\cdots ,2^{k-1}-1\right)$$
is satisfied for $i=2^{k-1},2^{k-1}+1,\cdots ,2^{k}-1$. Thus, we can write:
(30)$$\begin{array}{ccc} B_{i}^{\left(k\right)}\left(x\right)-\langle B_{i}^{\left(k\right)}\rangle & = & \sum_{j=0}^{2^{k-1}-1}\frac{1-h_{i,j}^{\left(k-1\right)}}{2}\left({\hat{\Theta }}_{\frac{j}{2^{k}}}\left(x\right)-{\hat{\Theta }}_{\frac{j+1}{2^{k}}}\left(x\right)\right)\\ & & +\sum_{j=0}^{2^{k-1}-1}\frac{1+h_{i,j}^{\left(k-1\right)}}{2}\left({\hat{\Theta }}_{\frac{j+2^{k-1}}{2^{k}}}\left(x\right)-{\hat{\Theta }}_{\frac{j+1+2^{k-1}}{2^{k}}}\left(x\right)\right). \end{array}$$
Further, for Bernoulli map, it is obvious that:
(31)$$\tau_{B}\left(x+\frac{1}{2}\right)=\tau_{B}\left(x\right)\;\mathrm{for}\;0\leq x<\frac{1}{2}.$$
Operating $P_{\tau_{B}}$ in Equation (Equation 30) and using Equations (Equation 28) and (Equation 31), we have:
(32)$$\begin{array}{ccc} P_{\tau_{B}}\left\{B_{i}^{\left(k\right)}\left(x\right)-\langle B_{i}^{\left(k\right)}\rangle \right\} & = & \sum_{j=0}^{2^{k-1}-1}\frac{1-h_{i,j}^{\left(k-1\right)}}{2}\cdot \frac{1}{2}\left({\hat{\Theta }}_{\tau_{B}\left(\frac{j}{2^{k}}\right)}\left(x\right)-{\hat{\Theta }}_{\tau_{B}\left(\frac{j+1}{2^{k}}\right)}\left(x\right)\right)\\ & & +\sum_{j=0}^{2^{k-1}-1}\frac{1+h_{i,j}^{\left(k-1\right)}}{2}\cdot \frac{1}{2}\left({\hat{\Theta }}_{\tau_{B}\left(\frac{j}{2^{k}}+\frac{1}{2}\right)}\left(x\right)-{\hat{\Theta }}_{\tau_{B}\left(\frac{j+1}{2^{k}}+\frac{1}{2}\right)}\left(x\right)\right)\\ & = & \frac{1}{2}\sum_{j=0}^{2^{k-1}-1}\left({\hat{\Theta }}_{\tau_{B}\left(\frac{j}{2^{k}}\right)}\left(x\right)-{\hat{\Theta }}_{\tau_{B}\left(\frac{j+1}{2^{k}}\right)}\left(x\right)\right)\\ & = & \frac{1}{2}\left({\hat{\Theta }}_{\tau_{B}\left(0\right)}\left(x\right)-{\hat{\Theta }}_{\tau_{B}\left(\frac{1}{2}\right)}\left(x\right)\right)\\ & = & 0, \end{array}$$
which completes the proof. □

**Remark** **5.**
*From Remarks 2 and 3 and the above Theorem, each of $2^{k-1}$ binary sequences ${\left\{B_{i}^{\left(k\right)}\left(\tau_{B}^{n}\left(x\right)\right)\right\}}_{n=0}^{\infty }$$\left(i=2^{k-1},2^{k-1}+1,\cdots ,2^{k}-1\right)$ is a balanced i.i.d. binary sequence, and they are uncorrelated (orthogonal) with each other for any time shift ℓ including ℓ=0, that is we have:*
(33)$$\begin{array}{ccc} C(\ell ;B_{i},B_{i}) & = & 0\;for\;\ell \geq 1, \end{array}$$
(34)$$\begin{array}{ccc} C(\ell ;B_{i},B_{j}) & = & 0\;for\;i\neq j,\;\ell \geq 0. \end{array}$$
*It should be noted that Equation (Equation 34) implies that $2^{k-1}$ binary sequences ${\left\{B_{i}^{\left(k\right)}\left(\tau_{B}^{n}\left(x\right)\right)\right\}}_{n=0}^{\infty }$ ($i=2^{k-1},2^{k-1}+1,\cdots ,2^{k}-1$) are essentially different, that is they are not time-shifted versions of the others. Table 1 shows the evaluation results for the case k=4.*


## 5. Conclusions

We theoretically evaluated the statistical properties of chaotic binary sequences generated by the Bernoulli map and Walsh functions. For given *k*, it was shown that $2^{k-1}$ binary sequences ${\left\{B_{i}^{\left(k\right)}\left(\tau_{B}^{n}\left(x\right)\right)\right\}}_{n=0}^{\infty }$
$\left(i=2^{k-1},2^{k-1}+1,\cdots ,2^{k}-1\right)$ are essentially different in the sense that none of them are time-shifted versions of the others. Furthermore, we showed that each of the $2^{k-1}$ binary sequences is a balanced *i.i.d.* sequence, and they are uncorrelated (orthogonal) with each other for any time shift.

As in [12,19], the Bernoulli map can be approximated by nonlinear feedback shift registers (NFSRs) [20] with finite bits, and the binary functions corresponding to $B_{i}^{\left(k\right)}\left(x\right)$ can be easily realized by combinational logic circuits. We will discuss the applications of the orthogonal binary sequences using such NFSRs in our future study.

## Figures and Tables

**Figure 1 entropy-21-00930-f001:**
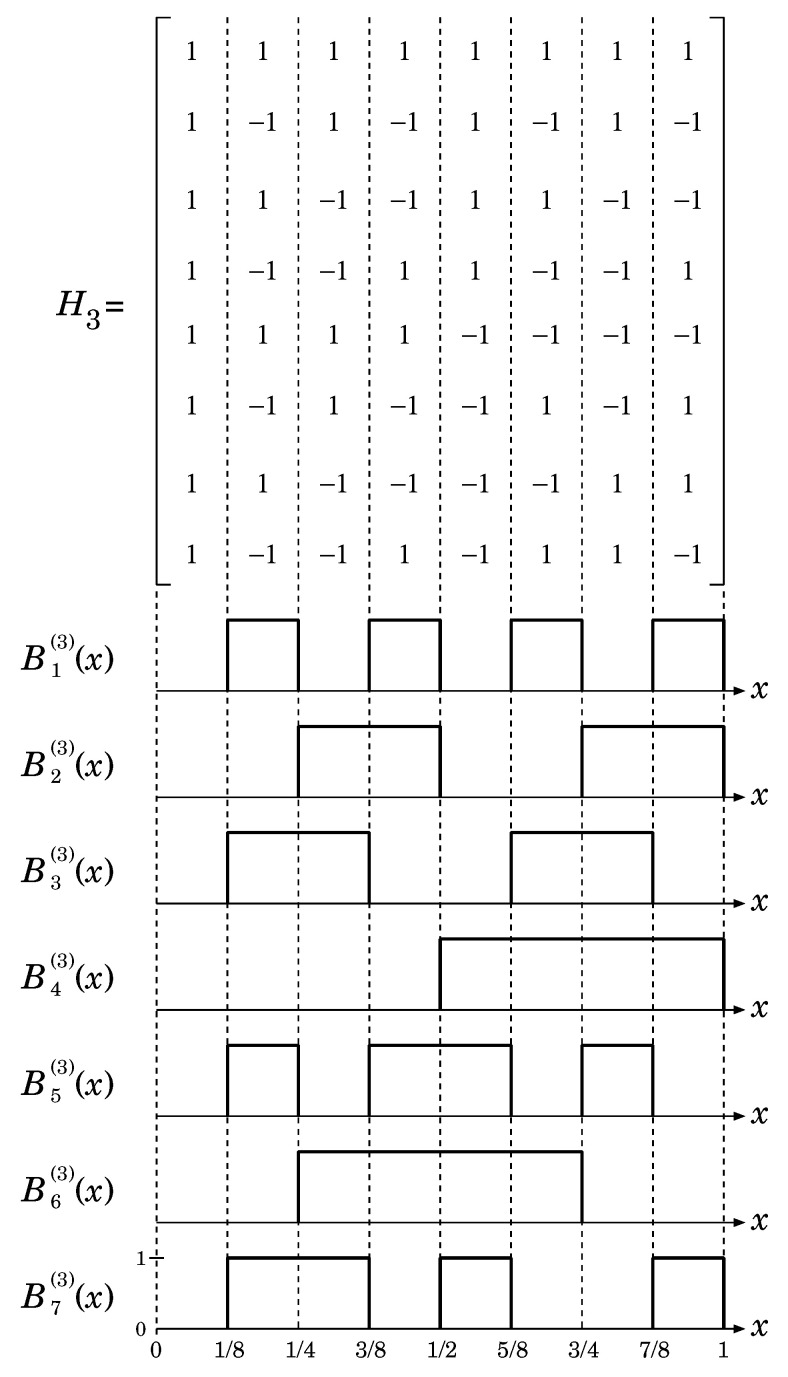
Binary (Walsh) functions based on an Hadamard matrix (k=3).

**Figure 2 entropy-21-00930-f002:**
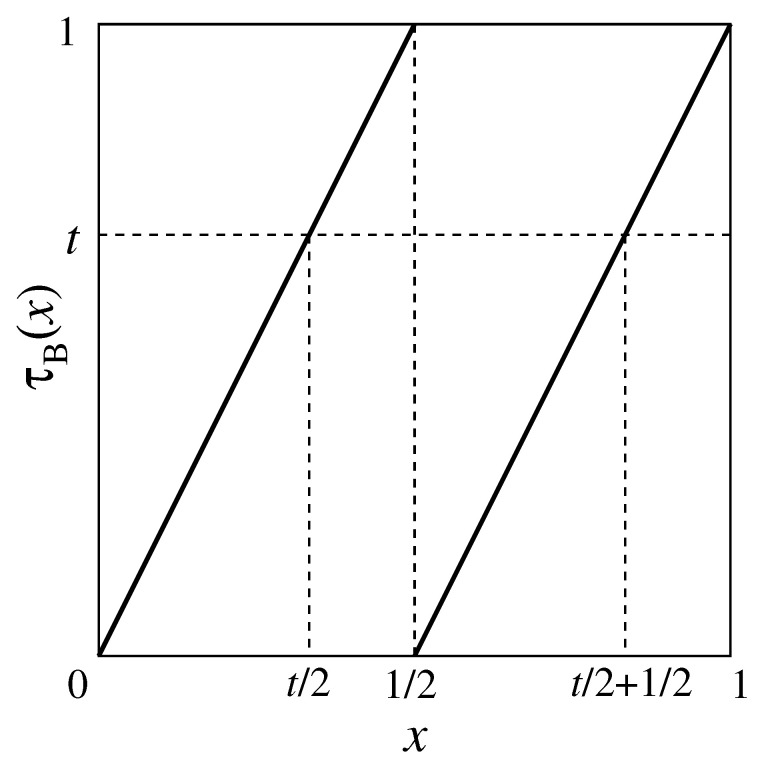
Bernoulli map.

**Table 1 entropy-21-00930-t001:** Evaluation results for the case k=4.

Binary Function	Evaluation Result
$B_{1}^{\left(4\right)}\left(x\right)$	$=B_{2}^{\left(4\right)}\left(\tau_{B}\left(x\right)\right)=B_{4}^{\left(4\right)}\left(\tau_{B}^{2}\left(x\right)\right)=B_{8}^{\left(4\right)}\left(\tau_{B}^{3}\left(x\right)\right)$
$B_{2}^{\left(4\right)}\left(x\right)=B_{1}^{\left(3\right)}\left(x\right)$	$=B_{4}^{\left(4\right)}\left(\tau_{B}\left(x\right)\right)=B_{8}^{\left(4\right)}\left(\tau_{B}^{2}\left(x\right)\right)$
$B_{3}^{\left(4\right)}\left(x\right)$	$=B_{6}^{\left(4\right)}\left(\tau_{B}\left(x\right)\right)=B_{12}^{\left(4\right)}\left(\tau_{B}^{2}\left(x\right)\right)$
$B_{4}^{\left(4\right)}\left(x\right)=B_{2}^{\left(3\right)}\left(x\right)$	$=B_{8}^{\left(4\right)}\left(\tau_{B}\left(x\right)\right)$
$B_{5}^{\left(4\right)}\left(x\right)$	$=B_{10}^{\left(4\right)}\left(\tau_{B}\left(x\right)\right)$
$B_{6}^{\left(4\right)}\left(x\right)=B_{3}^{\left(3\right)}\left(x\right)$	$=B_{12}^{\left(4\right)}\left(\tau_{B}\left(x\right)\right)$
$B_{7}^{\left(4\right)}\left(x\right)$	$=B_{14}^{\left(4\right)}\left(\tau_{B}\left(x\right)\right)$
$B_{8}^{\left(4\right)}\left(x\right)=B_{4}^{\left(3\right)}\left(x\right)$	balanced and *i.i.d.*
$B_{9}^{\left(4\right)}\left(x\right)$	balanced and *i.i.d.*
$B_{10}^{\left(4\right)}\left(x\right)=B_{5}^{\left(3\right)}\left(x\right)$	balanced and *i.i.d.*
$B_{11}^{\left(4\right)}\left(x\right)$	balanced and *i.i.d.*
$B_{12}^{\left(4\right)}\left(x\right)=B_{6}^{\left(3\right)}\left(x\right)$	balanced and *i.i.d.*
$B_{13}^{\left(4\right)}\left(x\right)$	balanced and *i.i.d.*
$B_{14}^{\left(4\right)}\left(x\right)=B_{7}^{\left(3\right)}\left(x\right)$	balanced and *i.i.d.*
$B_{15}^{\left(4\right)}\left(x\right)$	balanced and *i.i.d.*

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
