# Peer review of "Orthogonal Chaotic Binary Sequences Based on Bernoulli Map and Walsh Functions"

_entropy, 2019, doi:10.3390/e21100930_

Round 1
Reviewer 1 Report
The paper concerns statistical properties of sequences obtained as the result of a deterministic transformation of chaotic orbits produced by Bernoulli map. The author has chosen the Walsh functions, based on a 2^kx2^k Hadamard matrix, as such transformation.
The main conclusion of the paper is the statement that obtained 2^(k-1) binary sequences are different, balanced, i.i.d. and uncorrelated for a given k. Consequently, a one chaotic orbit of Bernoulli map can be the source of 2^(k-1) different, balanced, i.i.d. and uncorrelated sequences. Let us notice, that such sequences can also be produced by Bernoulli map itself, i.e., without any additional transformation. When “0” is produced for x<1/2 and “1” is produced for x>1/2 the output sequence is a binary representation of initial value of Bernoulli map’s trajectory. By choosing independently different seeds, different and independent binary sequences can be produced. The number of sequences is uncountable, because the number of initial points is also uncountable. Unfortunately, utility of such method is none. The output sequence is known a priori, i.e., before its generation. The author of the manuscript eliminates this drawback with additional transformation based on Walsh functions. Even though generated sequences are not known a priori, the problem of prediction subsequent zeros and ones from already generated elements of the sequence is still unsolved. The author’s proof that a single chaotic orbit of Bernoulli map can be a source of 2^(k-1) different, i.i.d., unbiased and uncorrelated sequences seems to be correct. In the reviewer’s opinion this result is useful and worth publication.
The sentence “Statistical properties of chaotic binary sequences generated by Bernoulli map and Walsh functions are discussed” suggests that 2^(k-1) different sequences are chaotic. In the reviewer’s opinion this may not be true, because not all transformations of chaotic orbits yield chaotic sequences. The critical element for proving chaotic behavior are not statistical properties of sequences but positive value of topological entropy of the source of sequences (see e.g. L. S. Block and W. A, Coppel ‘Dynamics in One Dimension,’ Springer-Verlag, 1992, Sec. VIII).
Reviewer 2 Report
Minor revision (corrections to minor methodological errors and text editing)
Author Response
Thank you so much for carefully reviewing my paper and giving your valuable comments. I revised the paper based on the reviewers’ comments including spell check.
Reviewer 3 Report
The article is very nice, thank you for sharing it! The manuscript is clearly written in professional, unambiguous language. The statistical analysis is very good and well summarized. However,The most current reference is 14 years old. I suggest adding at least one reference under 10 years.
minor spell check required:
Line 37: "eqs.(3) and (6)" by "Equations (3) and (6)"
Line 44: "Fig.1." by "Figure 1."
Lines 48, 55 and 61: "eq.(14)" by "Equation (14)"
Line 56: "Fig.2. Using eqs.(15),(24),(25)" by "Figure 2. Using Equations (15), (24) and (25)"
Line 63: "eq.(30) and using eqs.(28), (31)" by "Equation (30) and using Equations (28) and (31)"
Line 69: "eq.(34)" by "Equation (34)"
